# A Case Series Focusing on Blunt Traumatic Diaphragm Injury at a Level 1 Trauma Center

**DOI:** 10.3390/biomedicines13020325

**Published:** 2025-01-30

**Authors:** Bharti Sharma, Musili Kafaru, George Agriantonis, Aden Davis, Navin D. Bhatia, Kate Twelker, Zahra Shafaee, Jasmine Dave, Juan Mestre, Jennifer Whittington

**Affiliations:** 1Trauma Unit, Department of Surgery, NYC Health & Hospitals/Elmhurst, Queens, NY 11373, USA; kafarum@nychhc.org (M.K.); agriantg@nychhc.org (G.A.); davisa64@nychhc.org (A.D.); bhatian1@nychhc.org (N.D.B.); twelkerk1@nychhc.org (K.T.); shafaeez1@nychhc.org (Z.S.); davej@nychhc.org (J.D.); mestreju@nychhc.org (J.M.); harrisj20@nychhc.org (J.W.); 2Trauma Unit, Department of Surgery, Icahn School of Medicine at Mount Sinai Hospital, New York, NY 10029, USA

**Keywords:** trauma, blunt, diaphragmatic injury, blunt traumatic diaphragm injury

## Abstract

**Introduction:** Detection of blunt traumatic diaphragm injury (TDI) can be challenging in the absence of surgical exploration. Our objective is to study the mechanisms of injury and detection modes for patients with blunt TDI. **Methods:** This is a single-center, retrospective review conducted in a level 1 trauma center from 2016 to 2023, inclusive. We identified seven patients with blunt TDI using the primary mechanisms and trauma type. **Results:** Out of seven patients, two were associated with motor vehicle collisions, four were pedestrians struck, and one fell down the stairs. The mean ISS was 48.4 (29–75). Of the seven patients with blunt TDI, four died in the trauma bay–two from traumatic arrest and two died spontaneously. Multiple rib fractures were one of the common injury patterns in six cases, whereas in the remaining case, blunt TDI was confirmed at laparotomy and repaired. One patient died two days after admission. Of the two patients who survived, one had a TDI identified during video-assisted thoracic surgery (VATS) for retained hemothorax, and one patient had a TDI repaired during emergent exploratory laparotomy for other injuries. In the remaining four patients, blunt TDI was confirmed based on their autopsy reports. **Conclusions:** Injuries in all seven cases were sustained with a high-energy injury mechanism. Multiple rib fractures were reported in six cases. Based on our findings, we recommend that clinicians maintain a high level of suspicion for blunt TDI in patients with thoracoabdominal trauma, especially in cases with rib fractures or high-impact trauma.

## 1. Introduction

The diaphragm is a crucial skeletal muscle that separates the thoracic and abdominal cavities [1] and plays an essential role in activities like defecation and vomiting by regulating intra-thoracic and intra-abdominal pressure gradients [2]. Embryologically, it develops from the septum transversum during week four of gestation and assumes a domelike appearance with a central tendinous part upon development [2]. Anatomically, it attaches to the sternum, the sixth to twelfth ribs, the last thoracic vertebra, and the first three lumbar vertebrae [2]. Despite its protection within the thoracic cage, the diaphragm can sustain traumatic injuries, either blunt or penetrating injury, through the thoracoabdominal region. Blunt injuries often result from motor vehicle accidents, falls from a height, or pedestrian strikes, whereas penetrating injuries include gunshot wounds, stab wounds, or sharp objects with the capacity to penetrate the thoracoabdominal partition. Injury to the diaphragm can also arise at birth, a term called congenital diaphragmatic hernia, which occurs due to a defect in the diaphragm during its development [3].

Diagnosing a blunt traumatic diaphragm injury (TDI) can be challenging because of its potential to remain undetected in a few cases, leading to delayed presentation with abdominal pain and respiratory symptoms several months after the acute event [4]. Early diagnosis is crucial to prevent complications such as strangulation and perforation, which can arise if the injury is not promptly managed [5,6]. Chest X-ray (CXR) and contrast-enhanced computed tomography (CT) are the primary imaging modalities for diagnosing blunt TDI [7]. While CXR is often the first imaging technique used in trauma patients, studies show that it is diagnostic in only about one-third of cases [8].

On the other hand, CT is more sensitive and specific, detecting DI with a sensitivity of 71% and a specificity of 100%, particularly when abdominal contents are herniated into the thoracic cavity [8,9]. CT also provides detailed visualization of air or fluid in the thoracic cavity, helping to identify ruptures in the diaphragm [9]. Despite its higher sensitivity, CT is not infallible, and TDI may still be overlooked, especially when the injury is minute or when other associated injuries are more clinically apparent [10]. In cases focusing on left-sided blunt diaphragmatic ruptures, preoperative identification has been reported to occur in only 40–50% of cases and in a mere 0–10% for right-sided ruptures. Therefore, while CT is valuable, it may still require confirmation through surgical exploration, such as laparotomy or thoracotomy, in cases where clinical suspicion remains high.

Blunt TDI is uncommon and there are not many reported studies focusing on blunt TDI. Hence, to analyze various methods of detecting blunt TDI, the mechanism of injury, and the role of the Injury Severity Score (ISS) in patient outcomes, we bring forth this case series. We also reviewed patient demographics. This case series included seven patients from 2016 to 2023 with blunt TDI. The demographic breakdown included one White American, one Black American, two Asian patients, and three patients from other racial backgrounds, with a mean age of 47.1 years (range 28–88 years). Further details on all included cases are elaborated in the following sections.

## 2. Methods

This is a single-center, retrospective review conducted at a level 1 trauma center verified by the American College of Surgeons in Queens, New York City. We included all patients who presented with a blunt TDI from 1 January 2016 to 31 December 2023, inclusive. Patients who did not sustain injuries in blunt TDI (for example penetrative trauma) were excluded. The medical charts of the patients were reviewed, and all relevant information available for this study was collected.

Patients were selected from the National Trauma Registry of the American College of Surgeons (NTRACS) Database based on the injury mechanism, type of trauma, primary mechanisms (lCD9 or lCDL0 E-Code) or Injury code descriptions, and the Abbreviated Injury Severity (AIS) score body region classifications. The AIS score ranges from 1 to 6 per body region.

Based on trauma registry records, we identified 16 cases with blunt traumatic injuries. Out of these 16, the majority of patients were dead. Patients with maximum descriptive details were included in this case series, i.e., 7 cases. For patients who passed away without imaging records, blunt TDI was confirmed utilizing the trauma registry records. At our center, we update the trauma registry based on the autopsy results. Therefore, blunt TDI was confirmed for patients who died on arrival or who arrived already deceased.

We collected data using a data collection tool (Excel sheet or spreadsheet). We incorporated all data elements into this tool. Examples of data elements are demographics; AIS, ISS, and Glasgow Coma Scale (GCS); pre-existing conditions; patient vitals; procedures; ICU, ED, and hospital length of stay; ventilator days; mortality or discharge status; and others. These data (wherever applicable) were used to write this case series. We collected data using electronic medical records and requested relevant information from the trauma registry. For patients who died on arrival or arrived dead, autopsy reports were requested from the trauma registrar. This was essential to confirm the presence of blunt TDI in the patients who did not survive. Although autopsy is a surgical procedure, autopsy results are only updated in the registry at our facility.

In this case series, we present the cases of seven patients with blunt TDI. We include details about their demographics, mechanisms of injury, methods of detection, Injury Severity Scores (ISSs), clinical outcomes, and imaging results, when applicable.

## 3. Case Presentation

### 3.1. Case 1

An 88-year-old White American woman with a past medical history of hypertension, hyperlipidemia, and hip fracture with repair (4 years ago) presented to our emergency department (ED) with complaints of pain in her right forearm, left hand, lower back, and on the chest bilaterally with trouble breathing. According to the patient, she had fallen at home 2 days prior. She did not remember how she fell or the precise location but believed she tripped. The patient admitted to an extensive smoking history and likely had undiagnosed chronic obstructive pulmonary disease (COPD). She denied any loss of consciousness, dizziness, or hitting her head. The electrocardiogram (EKG) finding was significant for rapid atrial fibrillation (no prior history) which was managed with an anti-arrhythmic medication throughout her hospital course. The patient was tachycardic, tachypneic, hypotensive with blood pressure (BP) of 96/62 mmHg, and hypoxic to 88% for which she was placed on supplemental oxygen but afebrile in the ED. Physical examination noted tenderness to the right lateral chest with bruising to the right breast and right forearm with an overlying laceration. The abdomen was soft and non-tender. There was no sign of labored breathing. Chest X-ray (CXR) revealed right pleural effusion and areas of atelectasis with several right and lower fractured ribs. Contrast-enhanced computed tomography (CT) of the chest (Figure 1A) showed a perforated right hemidiaphragm with gastrointestinal content in the thorax, mildly displaced acute fractures anterior right third, fourth, and fifth ribs, mildly displaced acute fractures of the right eighth to tenth ribs, and markedly displaced acute fractures of the right sixth to seventh ribs. CT abdomen and pelvis revealed no acute findings. The patient refused surgical intervention. Two days after admission to the surgical intensive care unit (SICU), the patient experienced respiratory distress with the use of accessory muscles to breathe and became hypotensive for which she responded to fluid resuscitation. CT angiogram (CTA) was performed which showed herniation of the bowel into the right anterior chest significantly compromising the right lung volume, pleural effusion, and small pneumothorax (Figure 1B). The patient again refused surgical management and signed a do not intubate and do not resuscitate (DNI/DNR) order. The patient requested comfort care only and died the next day because of her condition. The Injury Severity Score (ISS) recorded for this patient was 29.

### 3.2. Case 2

A 30-year-old non-Hispanic male with no past medical history was brought to the emergency department (ED) by the emergency medical services (EMS) after being involved in a motor vehicle accident. The patient was riding his motorcycle when he got struck on his left side by a car traveling at 30 miles per hour (mph). Per the patient, he had a helmet on which remained intact after the collision. He denied losing consciousness but was not ambulating at the scene. The patient was hemodynamically stable at the time of presentation with a BP of 110/77 mmHg. The primary survey was remarkable for mild crepitus on the left lateral chest, a major deep laceration on the left chin, a deep wound on the right knee, and an abrasion to the left flank. Extended Focused Assessment with Sonography in Trauma (eFAST) was negative for any pericardial fluid. CXR showed a rib fracture on the left without pneumothorax. CT chest revealed mildly displaced fractures of the left 6th through 12th ribs posteriorly, contusion along the left lateral chest wall, and subcutaneous emphysema. CT abdomen and pelvis (CTAP) showed grade 2 or 3 splenic laceration with a small amount of perisplenic fluid for which the patient underwent embolization of the splenic artery. Due to CT chest findings of traumatic fractures of left 6th–12th ribs, the patient was taken to the operating room for left video-assisted thoracic surgery (VATS), left thoracotomy, and plating of left 7th–10th ribs. During thoracotomy exploration, it was discovered that the 10th rib had torn the diaphragm, and this was repaired with 0-ethibond running stitch and a 28 French chest tube was placed on the left. The patient was discharged on postoperative day 4 with pain medication and in stable condition. The reported ISS was 22.

### 3.3. Case 3

A 28-year-old Asian female with an unknown past medical history was brought by EMS as a pedestrian struck vs. jumped in front of a car at a high rate of speed. She was thrown onto her left side. The initial GCS was 10. On arrival at the ED, the patient was spontaneously moving but mumbling incoherently. Vitals were remarkable for tachycardia, tachypneic, and hypoxia. BP was 136/107 mmHg. The primary survey was notable for deformity of the left forearm and dilated unreactive left pupil. A portable pelvic ultrasound revealed a pelvic fracture. Chest ultrasound showed elevation of the left hemidiaphragm and comminuted diaphyseal fracture of the left humerus with major fragments. CTA of the neck showed C5-C6 with 3 mm posterior subluxation, irregular narrowing of the right vertebral artery extending from C5 to 7, and nondisplaced coracoid fracture of the left scapula. CT chest with contrast (Figure 2) revealed a distended stomach with air fluid in the left hemithorax concerning a closed loop obstruction. eFAST was negative for pericardial fluid. CTAP was consistent with CT chest findings including grade 4 splenic injury with contrast extravasation, grade 3 left renal injury with perinephric collection, and multiple pelvic fractures. The patient became hemodynamically unstable and was emergently taken to the operating room (OR) for exploratory laparotomy. The stomach was successfully reduced into the abdominal cavity from the thorax and on exploration, the diaphragm was noted to have a 6 cm defect, which was repaired with 2-0 ethibond in interrupted fashion with a total of 8 stitches. The patient also underwent splenectomy due to pulsatile red blood extravasation. On the same admission, neurosurgery performed anterior cervical discectomy and fusion and open reduction internal fixation (ORIF) of open humerus fracture and pelvic fracture. Two days after admission, she was taken back to the OR for repair of liver capsular laceration and facial closure. The patient was discharged on postoperative day 47 in stable condition. CXR and CT-A/P were consistent with left TDI. Hence, in this case, blunt TDI was confirmed at laparotomy and repaired. The recorded ISS was 41.

### 3.4. Case 4

A 36-year-old Hispanic female was brought by EMS after a pedestrian struck her. Initially, she had pulses on the scene but lost en route to the ED with 6 min of Cardiopulmonary Resuscitation (CPR) pre-hospital. Vitals were not accessible in the ED. Physical examination noted abrasion to the face, left leg deformity, and distended abdomen. The patient was intubated upon arrival and resuscitative thoracotomy was performed with no blood return; numerous rib fractures and contused lungs were noted, the diaphragm was significantly distended, and there was minimal cardiac activity. GCS was 3. There was no return of spontaneous circulation and the patient was pronounced dead in the ED. We requested patient data from the NTRACS Trauma Registry at our center. Based on autopsy results, the trauma registry records were updated. These records confirmed that the patient had a blunt diaphragmatic injury. As per the autopsy report, the recorded location of rib fractures was right-sided ribs 3–7, right lateral ribs 3–5, right paravertebral ribs 10–12, and left anterior ribs 2–8. The ISS reported for this patient was 43.

### 3.5. Case 5

A 68-year-old Asian female was brought in by EMS for traumatic arrest. The patient was struck by a motor vehicle on city streets. According to EMS, the patient was unresponsive and pulseless upon their arrival. The patient returned a pulse after a couple of rounds of CPR. Upon arrival at our emergency room, she was bagged with a bag valve mask but no response. Initial vitals were suggestive of hypovolemic shock with the absence of a pulse. Portable CXR showed multiple right-sided acute rib fractures and the possibility of impacted distal left clavicular fracture without pneumothorax. Portable pelvis ultrasound revealed a dislocated right femoral head. The patient did not respond to resuscitative efforts and was pronounced dead in the ED. Data for this patient were requested from the NTRACS Trauma Registry at our center. Based on autopsy results, the trauma registry records were updated. These records confirmed that the patient had a blunt diaphragmatic injury. As per the autopsy report, the location of the rib fractures was recorded as right-sided ribs 4, 6–8, and left lateral ribs 1–7. The patient’s recorded Injury Severity Score (ISS) was 54.

### 3.6. Case 6

A 39-year-old Hispanic brown female with an unknown past medical history was brought by EMS as a pedestrian struck by a motor vehicle. It was RED trauma on prenotification with a reported GCS of 15. She showed loss of vital signs within 6 min of prehospital CPR. She was intubated in the trauma bay. There was no detected End-tidal carbon dioxide (EtCO2), presumably due to lack of circulation. There was a chest tube extending through the right sixth intercostal muscles and into the right pleural space. Associated with chest tube placement was an approximately 1-1/2″ minimally hemorrhagic incision through the right hemidiaphragm and into the right lobe of the liver. MTP was activated before the patient’s arrival and resuscitative thoracotomy was performed. We requested patient data from the NTRACS Trauma Registry. At our center, the trauma registry records are updated based on autopsy results. The diaphragm was significantly distended reporting intra-abdominal hemorrhage. Autopsy results confirmed that the patient had a blunt diaphragmatic injury along with multiple hemorrhagic fractures of the left lateral ribs 1–7 and anterior right rib 4 and 6–8. The reported ISS for this patient was 75.

### 3.7. Case 7

A 32-year-old Black male was brought by EMS in traumatic arrest. The patient was pinned between the subway platform and the oncoming train. He suffered massive crush injuries to his torso, neck, and face. The patient was in cardiac arrest at the time of the paramedic’s arrival on the scene. Paramedics attempted to intubate but were unable to due to neck trauma. The patient was asystole on the scene and arrived at our ED without a pulse. Physical examination noted gross deformity to the right arm, right shoulder, pelvis, and ribs. There was no cardiac activity on ultrasound (US) and the patient was pronounced dead in the ED. No scans were obtained before death. The patient’s data were requested from the NTRACS Trauma Registry at our center. Based on autopsy results, the trauma registry records were updated. These records confirmed that the patient had a blunt diaphragmatic injury. As per the autopsy report, there was displaced fracture of all anterior ribs bilaterally, lateral left ribs 2–12, posterior left ribs 2–12, and posterior lateral right ribs 2–12. The recorded ISS for this patient was 75.

## 4. Management and Outcome

Diagnosing DI after a blunt trauma can be challenging due to its potential to remain undetected, leading to delayed presentation years after the initial injury [11,12]. The management of blunt TDI generally involves surgical intervention, with options including thoracotomy, laparotomy, or a combination of both, depending on the nature and severity of the injury. In stable patients with suspected TDI, diagnostic laparoscopy or thoracoscopy may be used to confirm the diagnosis [13,14,15,16,17]. For unstable patients or those with significant bowel herniation, immediate surgical exploration is warranted to prevent further complications [4]. The mortality rate related to DI can vary, ranging from 5% to 30% [15,16,17]. Right-sided TDI, while less common, carries an increased risk of mortality due to its association with other abdominal injuries [18]. Studies have shown that early surgical intervention, particularly in younger patients, is associated with better survival rates [12,18].

In our study, case 1 had blunt TDI detected on a chest CT scan, but the patient declined surgical intervention opting for comfort care and passing away the following day. This finding is consistent with the literature, indicating that right hemidiaphragm injury has an increased risk of mortality [18]. However, it is important to note that this case alone should not be used to generalize mortality rates for all patients with blunt TDI. The mortality observed in case 1 is an isolated instance and cannot be considered representative of the broader outcomes in the series.

In case 2, the patient underwent video-assisted thoracotomy surgery (VATS) for rib repair, and a DI was noted on the left, which was repaired with a 0-ethibond stitch, leading to a favorable outcome.

In case 3, blunt TDI was detected on exploratory laparotomy with repair using a 2-0 ethibond suture. The patient survived.

In case 4, resuscitative thoracotomy in the trauma bay revealed a distended diaphragm, but the patient succumbed to his injuries.

Cases 5, 6, and 7 died from traumatic arrest either before arriving at the ED or were pronounced dead in the trauma bay without any CT imaging performed to indicate DI before their death.

## 5. Discussion

One of the key findings of our study is the association between lower rib fractures (6th–12th ribs) and high-energy trauma as significant indicators of blunt TDI. In our cohort, three out of seven patients had fractures to the lower ribs, and all but one sustained high-energy trauma (pedestrian strike), supporting the role of these factors in increasing the suspicion for DI.

However, while our findings suggest a potential association, the small sample size limits the ability to make definitive conclusions, and further research with larger cohorts is needed to confirm these trends. Additionally, although radiographic imaging was used in many of our cases, we found that CXR and CT were not always sufficient in diagnosing TDI, particularly in cases where no bowel contents were herniating into the thoracic cavity. This supports findings from previous studies indicating that alternative diagnostic methods, such as diagnostic laparoscopy or surgical exploration, may be necessary in patients with high clinical suspicion but negative initial imaging [11].

Regarding the ISS, while our case series suggests that patients with ISS greater than 24 tend to have poorer outcomes, it is important to acknowledge the limitations of our small sample size. A retrospective study by Cardoso et al. found similar findings, with mortality rates higher in patients with an ISS greater than 24 [19], but larger, multicenter studies are needed to validate the predictive value of ISS for TDI outcomes.

Our findings emphasize the need for a high level of suspicion in cases of blunt trauma to the thoracoabdominal region, particularly in patients where CT does not demonstrate DI. These findings align with a retrospective study conducted on 155 patients at a level 1 trauma center [20], which also reported delays in diagnosing DI on initial imaging when there is no sign of DI. In that study, chest X-ray (CXR) was diagnostic in 21.4% of cases, while CT was diagnostic in 50.8%. However, 37.4% of diagnoses were made intraoperatively, often due to persistent pleural effusions and the presence of an elevated diaphragm, prompting further investigation. Delayed diagnosis of traumatic injuries like blunt TDI or others like traumatic brain injury (TBI) can result in mortality [20,21], particularly after 7 days, often occurring in patients with no herniated GI contents on imaging [22].

The studies discussed above can be co-related with our findings, indicating that a high level of suspicion is necessary for diagnosing TDI, particularly when no evident herniated contents are visible on CXR or CT. These insights should be incorporated into trauma protocols, emphasizing the importance of clinical judgment and the use of alternative diagnostic methods—such as diagnostic laparoscopy—in stable patients with suspicious signs but negative initial imaging. Given that our study found a similar diagnostic challenge, we propose that clinicians maintain a high threshold of suspicion for DI and consider more comprehensive diagnostic approaches, particularly in cases of high-energy trauma or when imaging findings are inconclusive.

We summarize various findings of the patients in this case series in Table 1. We explain the common trends and diagnostic challenges in these reports. The patients in this case series ranged from 28 to 88 years old, indicating that diaphragmatic injuries can affect both younger and older individuals. However, elderly patients are more likely to decline surgical intervention, which may lead to worsening outcomes. Many studies have found that blunt TDI is more common in males than in females, but in our case, the incidence was higher in females (62.5%) compared to males (37.5%). We noted no gender-specific trends in injury or outcomes. The CT scan’s inability to detect diaphragmatic injuries posed a significant diagnostic challenge, especially in cases where no visceral organs were herniating into the thoracic cavity. Another common trend is the mechanism of injury. Cases that involved pedestrian strikes or high-energy trauma had poor outcomes. Additionally, patients with lower Injury Severity Score (ISS) have favorable outcomes. A favorable outcome is also seen in one of our patients with higher ISS (>24) due to early surgical intervention. This supports the idea that timely and aggressive management can lead to good outcomes, even in cases with significant trauma.

As most of the cases in our study died on arrival or arrived dead, we reviewed autopsy reports to confirm blunt TDI in such cases. We only included cases with more detailed information so that the severity of blunt TDI can be communicated to other researchers. This ensures the novelty of our study. Among the several published papers, very few groups have focused on writing a case series solely focusing on blunt TDI. Archives of published studies either include a majority of case reports with single cases or very few clinical trials. Not only did we refer to the patient’s medical chart but we also referred to registry data to ensure exploration and detailed description of these cases wherever possible. At our center, out of 17 cases, except for a few, the rest were either dead on arrival or arrived dead due to injury severity or died within 2 days of admission. Our study is one of the novel studies proving the high mortality rate in patients with blunt TDI. Although the size of the studied patient population is small, it highlights the need to advance education programs to manage such life-changing incidents. The findings will impact trauma quality improvement practices regionally and nationally. Such crucial points impart novelty, educational value, and uniqueness to our study when compared to other reports.

## 6. Conclusions

In conclusion, our study highlights the importance of high suspicion for DI in cases with rib fractures from the 6th to 12th ribs or high-energy trauma. Patients with ISS greater than 24 had poorer outcomes regardless of the mode of mechanism, except in one case where the patient survived, likely due to timely operative intervention. CT radiography helped detect DI in a few cases but its failure to detect a DI in one patient highlights the need for timely surgical management, particularly in the setting of high clinical suspicion or severe ISS [5,6,9,12,20].

Based on our findings, we recommend that clinicians maintain a high level of suspicion for TDI in patients with thoracoabdominal trauma, especially in cases with rib fractures or high-impact trauma. Specifically, (1) for patients with mechanical falls, if they present with respiratory distress, a history of lung disease, and physical signs of chest wall trauma, we recommend prompt laparotomy or diagnostic laparoscopy, depending on hemodynamic stability. (2) For pedestrian or motor vehicle trauma, urgent surgical intervention should be considered for hemodynamically unstable patients with suspected DI. If imaging fails to show clear signs of TDI, diagnostic laparoscopy or thoracoscopy should be performed, with serial imaging and follow-up to ensure no delayed injuries are missed in stable patients.

These recommendations should be viewed as preliminary suggestions based on our case series and not definitive treatment strategies, given the small sample size and the observational nature of this study.

Given the limitations of our study, including its small sample size, we recommend larger, multicenter studies to further evaluate the optimal imaging and surgical approaches for TDI diagnosis. Research should focus on the development of imaging protocols to enhance the sensitivity of CT scans, particularly in cases without visible bowel herniation, and on establishing protocols for post-discharge monitoring of patients at risk for delayed TDI diagnosis.

## Figures and Tables

**Figure 1 biomedicines-13-00325-f001:**
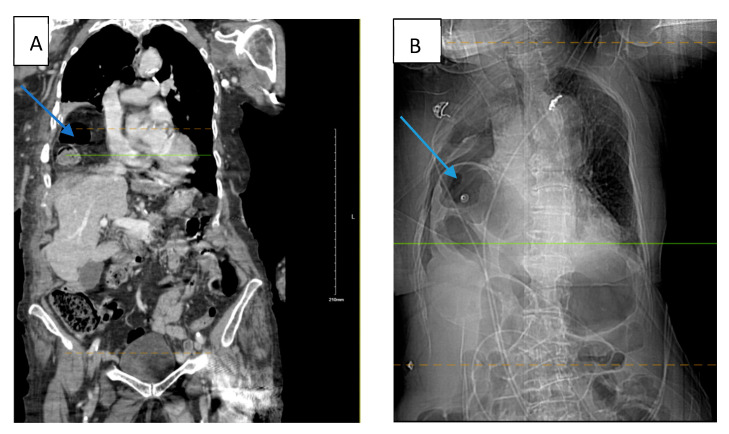
(**A**) Contrast-enhanced computed tomography (CT) (at the time of admission) of the chest showing a large right diaphragmatic hernia containing hepatic flexure of the colon and omental fat without evidence of strangulation (blue arrow). (**B**) CTA (two days after admission) revealed a large intestine in the right chest consistent with worsening herniation with the right lung completely collapsed by the herniated colon. Blue arrow in both figures are pointing towards the injury.

**Figure 2 biomedicines-13-00325-f002:**
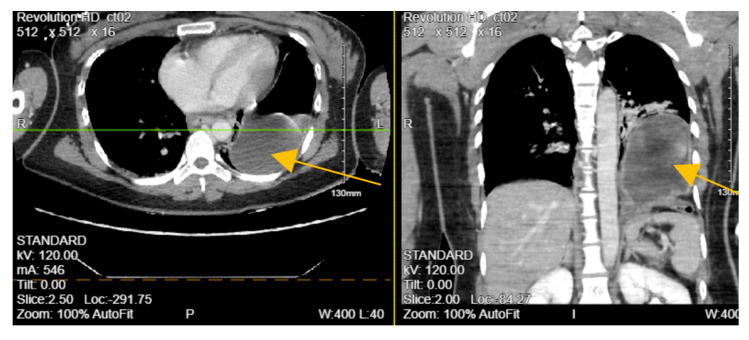
CT of the chest with contrast showing distended stomach with air and fluid in the left hemithorax (orange arrow), diaphragmatic defect, and grade IV splenic injury parenchymal laceration greater than 3 cm.

**Table 1 biomedicines-13-00325-t001:** Summary of all cases with Blunt Traumatic Diaphragmatic Injury (TDI) discussed in this case series.

Case No.	Mechanism of Injury	Patient Demographics	Age	Gender	CT Scan Findings	Surgical Intervention	Outcome	ISS	Location of Rib Fracture
1	Mechanical fall at home	White American	88	Female	Abdominal contents in right hemidiaphragm	Comfort care (no surgery)	Died next day	29	anterior right 3–5 ribs, right 8–10 ribs, and right 6–7 ribs
2	Motor vehicle accident	non-Hispanic	30	Male	Mildly displaced fracture of left ribs 6th–12th	VATS and thoracotomy for rib repair, diaphragm repair	survived	22	left 6 –12 ribs
3	Pedestrian strike	Asian	28	Female	Abdominal content in left hemithorax	Exploratory laparotomy, diaphragm repair	survived	41	No rib fracture location; blunt TDI was confirmed at laparotomy and repaired
4	Pedestrian strike	Hispanic	36	Female	Elevated diaphragm (no CT scan)	Resuscitative thoracotomy	died	43	right-sided ribs 3–7, right lateral ribs 3–5, right paravertebral ribs 10–12, and left anterior ribs 2–8
5	Pedestrian strike	Asian	68	Female	No CT scan performed	No surgical intervention	died	54	right-sided ribs 4, 6–8 and left lateral ribs 1–7
6	Motor vehicle accident	Hispanic	48	Male	No CT scan performed	No surgical intervention	Died	75	multiple hemorrhagic fractures of the left lateral ribs 1–7 and anterior right rib 4 and 6–8.
7	Pedestrian strike	Black American	32	Male	No CT scan performed	Died without surgical intervention	died	75	all anterior ribs bilaterally, lateral left ribs 2–12, posterior left ribs 2–12, and posterior lateral right ribs 2–12

## Data Availability

The original contributions presented in the study are included in the article, further inquiries can be directed to the corresponding author.

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
