# Peer review of "A Case Series Focusing on Blunt Traumatic Diaphragm Injury at a Level 1 Trauma Center"

_biomedicines, 2025, doi:10.3390/biomedicines13020325_

Round 1

Reviewer 1 Report

Comments and Suggestions for Authors

The article, A Case Series Focusing on Blunt Traumatic Diaphragm Injury at a Level 1 Trauma Center, highlights an important yet underreported injury type in trauma care. However, it requires significant revisions to improve clarity and relevance. The introduction should include more background on blunt traumatic diaphragm injuries (BTDI), outlining epidemiology, mechanisms, and diagnostic challenges to better contextualize the case series. The methodology lacks detail on case selection, diagnostic criteria, and data collection, making it hard to assess the rigor of the study. The results could be strengthened by a structured summary table of cases, and the analysis would benefit from highlighting common trends or diagnostic challenges across cases. In the discussion, the authors should compare their findings with existing BTDI research and offer insights on diagnostic issues or potential improvements in trauma protocols. A clearer conclusion with actionable recommendations for trauma clinicians, such as suggestions for diagnostic protocols and a call for further research, would enhance the paper’s practical impact. Overall, more comprehensive context, structured data, and practical recommendations are needed to make this case series a valuable resource.

Author Response

Greetings reviewer,

Thank you for the time taken to provide wonderful comments and feedback on this report. We hope our responses address your questions.

Reviewer 1: (All edits in manuscripts are highlighted in light blue)

  1. The introduction should include more background on blunt traumatic diaphragm injuries (BTDI), outlining epidemiology, mechanisms, and diagnostic challenges to better contextualize the case series.

Response:  We have added 3 paragraphs elaborating on BTDI at the end of the introduction section

Traumatic diaphragmatic injuries (TDI) are not common, with blunt TDI occurring in 1 to 7% of cases and penetrating injuries occurring in 15% of cases (2, 4). Patients may present with chest pain and respiratory symptoms, raising suspicion of DI. Diaphragmatic injuries are relatively rare after blunt trauma and are often missed diagnoses, leading to delayed surgery (5). Blunt traumatic diaphragmatic rupture can be due to an increase in kinetic energy transferred to the diaphragm (5). The left hemidiaphragm is more commonly affected due to its congenital weakness during development (5), whereas the right is less vulnerable to injury due to the liver acting as a protective mechanism. Although it is rare, right-sided DI carries an increased risk of mortality due to association with other abdominal injuries (6). Penetrating injuries typically follow the path of the foreign object and are relatively small in extent (2). Studies show that motor vehicle collisions are the most common cause of blunt TDI (63.4%), while gunshot wounds are the predominant cause of penetrating TDI (66.5%) (7).

The diaphragm plays a crucial role in normal breathing, and injuries to this muscle can lead to serious difficulties in ventilation. Diagnosing such injuries can be challenging; preoperative identification occurs in only 40-50% of cases involving left-sided blunt diaphragmatic ruptures and a mere 0-10% for right-sided ruptures. Furthermore, in 10-50% of patients, the diagnosis may not be established within the first 24 hours following the injury. Bilateral injuries are quite rare, occurring in approximately 3% of cases. It’s important to note that early fatalities are often due to associated injuries rather than the diaphragmatic rupture itself. A study analyzing data from the National Trauma Databank has revealed that blunt TDI constitutes only 0.46% of all trauma cases, with two-thirds of these as penetrating injuries. The mortality rate related to diaphragmatic injuries can vary, ranging from 5% to 30% (4, 8-11).

The process of diagnosis and treatment tends to be alike, irrespective of the underlying mechanism, although numerous management concerns are particular to blunt trauma (12). Chest X-rays are the most crucial diagnostic examination. They can reveal an elevated hemidiaphragm, a bowel pattern located in the chest, or a nasogastric (NG) tube that extends into the abdomen and then curves up into the chest. A chest X-ray of a blunt injury to the left diaphragm frequently displays an unusual or enlarged mediastinum, even when the aorta appears normal. The mediastinum should be further examined due to its connection with aortic injuries (13). CT scanning is a valuable diagnostic tool, though it has limitations, particularly in visualizing the diaphragm. A definitive diagnosis of diaphragmatic injury can be made if herniation of abdominal contents is observed. A study conducted by the Northern French Alps Emergency Network found that chest radiographs were only diagnostic in 18 out of 29 patients with blunt trauma diaphragmatic injuries among the 31 patients studied. In contrast, CT scans provided a diagnostic result for 26 out of the 29 patients. For stable trauma patients, the use of contrast-enhanced abdominothoracic CT along with reconstruction techniques led to earlier detection of these injuries (13,14). The diagnosis might be postponed or overlooked due to the ambiguous clinical and radiographic results and several related injuries (14, 15).

  1. The methodology lacks detail on case selection, diagnostic criteria, and data collection, making it hard to assess the rigor of the study.

Response: We have added a new section for methods

Methods

This is a single-center, retrospective review conducted at a level 1 trauma center verified by the American College of Surgeons in Queens, New York City. We included all patients who presented with a blunt traumatic diaphragm injury (TDI) from January 1, 2016, to December 31, 2023, inclusive. Patients who did not sustain injuries in blunt TDI (for example penetrative trauma) were excluded. The medical charts of the patients were reviewed, and all relevant information available for this study was collected.

Patients were selected from the National Trauma Registry of the American College of Surgeons (NTRACS) Database based on the injury mechanism, primary mechanisms (lCD9 or lCDL0 E-Code), or Injury code descriptions, and the Abbreviated Injury Severity (AIS) score body region classifications. The AIS score ranges from 1 to 6 per body region.

Based on trauma registry records we identified 16 cases with blunt traumatic injuries. Out of these 16, the majority of patients were dead. Patients with maximum descriptive details were included in this case series i.e. 8 cases. For patients who passed away without imaging records, blunt traumatic brain injury (TBI) was confirmed utilizing the trauma registry records. At our center, we update the trauma registry based on the autopsy results. Therefore, blunt TBI was confirmed for patients who died on arrival or who arrived already deceased.

We collected data using a data collection tool (Excel sheet or spreadsheet). We incorporated all data elements into this tool. Examples of data elements are demographics, AIS, ISS, and Glasgow Coma Scale (GCS), pre-existing conditions, patient vitals, procedures, ICU, ED, and hospital length of stay, ventilator days, mortality or discharge status, and others. This data (wherever applicable) was used to write this case series. We collected data using electronic medical records and requested relevant information from the trauma registry. For patients who died on arrival or arrived dead, autopsy reports were requested from the trauma registrar. This was essential to confirm the presence of Blunt TDI in the patients who couldn’t survive. Although autopsy is a surgical procedure autopsy results are only updated in the registry at our facility.

In this case series, we present the cases of eight patients with blunt traumatic dental injuries (TDI). We include details about their demographics, mechanisms of injury, methods of detection, injury severity scores (ISS), clinical outcomes, and imaging results, when applicable.

  1. The results could be strengthened by a structured summary table of cases, and the analysis would benefit from highlighting common trends or diagnostic challenges across cases.

Response: Thank you for highlighting this point. We have now modified the summary table and discussed the common trends and diagnostic challenges across the cases in the table legend in the discussion session.

Table 1: Summary of all cases with Blunt Traumatic Diaphragmatic Injury (TDI) discussed in this case series

Case No.

Mechanism of injury

Patient

demographics

Age

Gender

CT scan findings

Surgical intervention

Outcome

ISS

Comments

1

Mechanical fall at home

White American

88

Female

Abdominal contents in right hemidiaphragm

Comfort care (no surgery)

Died next day

29

Increased mortality with right hemidiaphragm injury

2

Motor vehicle accident

non-Hispanic

30

Male

Mildly displaced fracture of left ribs 6th – 12th

VATS and thoracotomy for rib repair, diaphragm repair

survived

22

CT missed injury; VATS revealed a diaphragmatic tear

3

Pedestrian strike

Asian

28

Female

Abdominal content in left hemithorax

Exploratory laparotomy, diaphragm repair

survived

41

Survived with surgical intervention

4

Mechanical fall at home

Hispanic

74

Female

Right hemidiaphragm elevation on noncontrast CT

Exploratory laparoscopy, no injury found

survived

9

Laparoscopy was effective in stable patients

5

Pedestrian strike

Hispanic

36

Female

Elevated diaphragm (no CT scan)

Resuscitative thoracotomy

died

43

High-impact injury, no survival

6

Pedestrian strike

Asian

68

Female

No CT scan performed

No surgical intervention

died

54

Died

7

Motor vehicle accident

Hispanic

48

Male

No CT scan performed

No surgical intervention

Died

75

Died

8

Pedestrian strike

Black American

32

Male

No CT scan performed

Died without surgical intervention

died

75

High-impact injury, no survival.

Table 1: We explained the common trend and diagnostic challenges in this case series. The patients in this case series ranged from 28 to 88 years old, indicating that diaphragmatic injuries can affect both younger and older individuals. However, elderly patients are more likely to decline surgical intervention, which may lead to worsening outcomes. Many studies have found that Blunt Traumatic Diaphragmatic Injury (BTDI) is more common in males than in females, but in our case, the incidence was higher in females (62.5%) compared to males (37.5%). We noted no gender-specific trends in injury or outcomes. The CT scan's inability to detect diaphragmatic injuries posed a significant diagnostic challenge, especially in cases where no visceral organs were herniating into the thoracic cavity. In case 2, where CT scans failed to reveal diaphragmatic rupture, surgical exploration (VATS or thoracotomy) ultimately revealed the injuries. In case 4, where CT findings are inconclusive or absent, diagnostic laparoscopy and resuscitative thoracotomy helped confirm or rule out diaphragmatic tears. Another common trend is the mechanism of injury. Cases 5, 6, and 8, all of which involved pedestrian strikes or high-energy trauma, had poor outcomes. Patients with lower Injury Severity Score (ISS) have favorable outcomes. A favorable outcome is also seen in one of our patients with higher ISS (>24) due to early surgical intervention. This supports the idea that timely and aggressive management can lead to good outcomes, even in cases with significant trauma.

  1. In the discussion, the authors should compare their findings with existing BTDI research and offer insights on diagnostic issues or potential improvements in trauma protocols.

Response: this has now been addressed in the last three paragraphs of the discussion session.

Overall, of our eight patients, three had CXR performed with one (33.3%) showing findings indicative of DI (case 3). Among the four patients who underwent CT imaging, DI was confirmed in two (50%) through the presence of herniated gastrointestinal (GI) content in the thorax (cases 1 and 3). However, CT failed to detect DI in the other two patients (cases 2 and 4), despite a high index of suspicion in case 4. This highlights that while CT is more sensitive and specific than chest radiographs, it may still miss injuries, especially when no visceral contents are present in the thoracic cavity, posing a challenge to early diagnosis.

Our findings emphasize the need for a high level of suspicion in cases of blunt trauma to the thoracoabdominal region, particularly in patients where CT does not demonstrate DI. These findings align with a retrospective study conducted on 155 patients at a level 1 trauma center (21), which also reported delays in diagnosing DI on initial imaging when no sign of DI. In their study, chest X-ray (CXR) was diagnostic in 21.4% of cases, while CT was diagnostic in 50.8%. However, 37.4% of diagnoses were made intraoperatively, often due to persistent pleural effusions and the presence of an elevated diaphragm, prompting further investigation. Delayed diagnosis of diaphragmatic injury, particularly after 7 days, often occurred in patients with no herniated GI contents on imaging (21).

Above discussed studies can be co-related with our findings indicating that a high level of suspicion is necessary for diagnosing TDI, particularly when no evident herniated contents are visible on CXR or CT. These insights should be incorporated into trauma protocols, emphasizing the importance of clinical judgment and the use of alternative diagnostic methods—such as diagnostic laparoscopy—in stable patients with suspicious signs but negative initial imaging. Given that our study found a similar diagnostic challenge, we propose that clinicians maintain a high threshold of suspicion for diaphragmatic injuries and consider more comprehensive diagnostic approaches, particularly in cases of high-energy trauma or when imaging findings are inconclusive

  1. A clearer conclusion with actionable recommendations for trauma clinicians, such as suggestions for diagnostic protocols and a call for further research, would enhance the paper’s practical impact.

Response: This has been addressed in the last three paragraphs of the conclusion section.

Based on our findings, we recommend that clinicians maintain a high level of suspicion for TDI in patients with thoracoabdominal trauma, especially in cases with rib fractures or high-impact trauma. Specifically: (1) for patients with mechanical falls: If they present with respiratory distress, a history of lung disease, and physical signs of chest wall trauma, we recommend prompt laparotomy or diagnostic laparoscopy, depending on hemodynamic stability. (2) For pedestrian or motor vehicle trauma: Urgent surgical intervention should be considered for hemodynamically unstable patients with suspected diaphragmatic injury. If imaging fails to show clear signs of TDI, diagnostic laparoscopy or thoracoscopy should be performed, with serial imaging and follow-up to ensure no delayed injuries are missed in stable patients.

We call for the development of standardized trauma protocols that incorporate these recommendations, particularly in settings with limited resources. Protocols should be based on the mechanism of injury and clinical presentation, physical exam findings, and radiographic imaging ensuring that patients with potential TDI receive timely intervention.

Given the limitations of our study, including its small sample size, we recommend larger, multicenter studies to further evaluate the optimal imaging and surgical approaches for TDI diagnosis. Research should focus on the development of imaging protocols to enhance the sensitivity of CT scans, particularly in cases without visible bowel herniation, and on establishing protocols for post-discharge monitoring of patients at risk for delayed TDI diagnosis.

Thanks again! We are willing to make any other necessary adjustments if needed. While an elevated diaphragm may not necessarily indicate a diaphragmatic injury, it can be a suggestive sign when considered in conjunction with other clinical signs and imaging findings.

Reviewer 2 Report

Comments and Suggestions for Authors

This is a case series report of traumatic diaphragmatic injuries at a single institution. Because traumatic diaphragmatic injury is rare, I believe that compiling it as a case series is a good choice, but there are the following major problems.

1.     Overall: The issues this report is trying to solve is not clear. Please clarify its novelty, educational value, and uniqueness compared to other reports. The current description is just a summary of past reports and does not convey the significance of this report.

2.     Methods: Please describe the characteristics of the facility where this study was conducted. It is unclear if it is described only as a level 1 trauma center. Please clarify the patient flow of nine cases selected from the total cases covered by the study from 2016 to 2023.

3.     Case Presentation: Case 3 was not a blunt trauma. It was a penetrated injury caused by a knife.

4.     Case Presentation: Diaphragmatic injury was not evident in cases 5, 7, 8, and 9. It is wondered why these cases were included in this study.

5.     Case Presentation: Case 1, please clarify which ribs were fractured. The position of the blue arrow in Figure 1A is incorrect. It is unclear what the CT image of the lung field condition on the right side of Figure 1A is trying to show. Please present images under the same conditions so that Figure 1B and Figure 1A can be compared.

6.     Case Presentation: Case 2, I'm not sure what additional meaning Figure 2 try to show in this report, which deals with traumatic diaphragmatic injury.

7.     Case Presentation: Case 4, Figure 3 is not an ultrasound image. It is unclear what the blue arrows indicate. I do not understand why a pelvic image is necessary in this report.

8.     Case Presentation: Case 7, Figure 4 is not an ultrasound image. It is unclear why a pelvic image is required.

9.     Discussion and Conclusion: It is inappropriate to consider the mortality rate based on the course of only one case. In addition, I cannot agree with the claim in Case 5 that diagnostic endoscopy should be performed if the patient has diaphragm elevation in a patient whose general condition is stable. Diaphragm elevation is due to a mechanism different from diaphragm injury.

Author Response

Greetings reviewer,

Thank you for the time taken to provide wonderful comments and feedback on this report. We hope our responses address your questions.

.

Reviewer 2: (All edits in manuscripts are highlighted in light blue)

  1. Overall: The issues this report is trying to solve is not clear. Please clarify its novelty, educational value, and uniqueness compared to other reports. The current description is just a summary of past reports and does not convey the significance of this report.

Response: We have added a new elaborative paragraph on novelty, educational value, and uniqueness at the end of the discussion section

As most of the cases in our study died on arrival or arrived dead, we reviewed autopsy reports to confirm blunt TDI in such cases. We only included cases with more detailed information so that the severity of blunt TDI can be communicated to other researchers. This ensures the novelty of our study. Among the several published papers, very few groups have focused on writing a case series solely focusing on blunt TDI. Archives of published studies either include the majority of case reports with single cases or very few clinical trials. Not only did we refer to the patient’s medical chart, but we also referred to registry data to ensure exploration and detailed description of these cases wherever possible. At our center, out of 17 cases, except for a few, the rest were either dead on arrival or arrived dead due to injury severity or died within 2 days of admission. Our study is one of the novel studies proving the high mortality rate in patients with blunt TDI. Although the size of the studied patient population is small, it highlights the need to advance education programs to manage such life-changing incidents. The findings will impact trauma quality improvement practices regionally and nationally. Such crucial points impart novelty, educational value, and uniqueness to our study when compared to other reports.

  1. Methods: Please describe the characteristics of the facility where this study was conducted. It is unclear if it is described only as a level 1 trauma center. Please clarify the patient flow of nine cases selected from the total cases covered by the study from 2016 to 2023.

Response:  We have added a new section for methods

Methods

This is a single-center, retrospective review conducted at a level 1 trauma center verified by the American College of Surgeons in Queens, New York City. We included all patients who presented with a blunt traumatic diaphragm injury (TDI) from January 1, 2016, to December 31, 2023, inclusive. Patients who did not sustain injuries in blunt TDI (for example penetrative trauma) were excluded. The medical charts of the patients were reviewed, and all relevant information available for this study was collected.

Patients were selected from the National Trauma Registry of the American College of Surgeons (NTRACS) Database based on the injury mechanism, primary mechanisms (lCD9 or lCDL0 E-Code), or Injury code descriptions, and the Abbreviated Injury Severity (AIS) score body region classifications. The AIS score ranges from 1 to 6 per body region.

Based on trauma registry records we identified 16 cases with blunt traumatic injuries. Out of these 16, the majority of patients were dead. Patients with maximum descriptive details were included in this case series i.e. 8 cases. For patients who passed away without imaging records, blunt traumatic brain injury (TBI) was confirmed utilizing the trauma registry records. At our center, we update the trauma registry based on the autopsy results. Therefore, blunt TBI was confirmed for patients who died on arrival or who arrived already deceased.

We collected data using a data collection tool (Excel sheet or spreadsheet). We incorporated all data elements into this tool. Examples of data elements are demographics, AIS, ISS, and Glasgow Coma Scale (GCS), pre-existing conditions, patient vitals, procedures, ICU, ED, and hospital length of stay, ventilator days, mortality or discharge status, and others. This data (wherever applicable) was used to write this case series. We collected data using electronic medical records and requested relevant information from the trauma registry. For patients who died on arrival or arrived dead, autopsy reports were requested from the trauma registrar. This was essential to confirm the presence of Blunt TDI in the patients who couldn’t survive. Although autopsy is a surgical procedure autopsy results are only updated in the registry at our facility.

In this case series, we present the cases of eight patients with blunt traumatic dental injuries (TDI). We include details about their demographics, mechanisms of injury, methods of detection, injury severity scores (ISS), clinical outcomes, and imaging results, when applicable.

  1. Case Presentation: Case 3 was not a blunt trauma. It was a penetrated injury caused by a knife.

Response: this case has now been entirely removed given that it was a penetrative injury (note: this has changed the numbering of our cases).

  1. Case Presentation: Diaphragmatic injury was not evident in cases 5, 7, 8, and 9. It is wondered why these cases were included in this study.

Response: While an elevated diaphragm may not necessarily indicate a diaphragmatic injury, it can be a suggestive sign when considered in conjunction with other clinical signs and imaging findings. In case 5 (now case 4), imaging showed an elevated diaphragm and given the patient's history along with the clinical symptoms, diaphragmatic injury was suspected. Because the patient was hemodynamically stable, a diagnostic injury was performed to rule out diaphragmatic injury. This approach aligns with our Reference #15 (Al-Jehani and El-Ghoneimy, 2011), who suggest that diagnostic laparoscopy can be useful in stable patient. However, diaphragmatic injury was not identified during this procedure. Case 5 was included to demonstrate that an elevated diaphragm may not always be indicative of diaphragmatic injury and provide a limitation of solely relying on imaging findings.

For case 7, 8, 9 (now 6, 7, 8), although no radiographic scan was performed before these patients death, but according to trauma registry, ICD9 or ICDL0 E-code, they have blunt diaphragmatic injury).

Also in case 7 (now case 6, the ultrasound figure has now been removed as the focus of the case is on diaphragmatic injury).

  1. Case Presentation: In case 1, please clarify which ribs were fractured. The position of the blue arrow in Figure 1A is incorrect. It is unclear what the CT image of the lung field condition on the right side of Figure 1A is trying to show. Please present images under the same conditions so that Figure 1B and Figure 1A can be compared.

Response: We have discussed the ribs that were fractured in-text of case 1 presentation. Similarly, Figure 1 A and B has been modified without the lung consolidation. All arrows have been fixed to point to herniated bowel contents in the thorax (1A) and worsening herniation (1B).

  1. Case Presentation: Case 2, I'm not sure what additional meaning Figure 2 tries to show in this report, which deals with traumatic diaphragmatic injury.

Response: Thank you for pointing this out. The Figure has now been removed as it has nothing to do with the focus of this report.  (Note: this has changed the numbering of our Figures).

  1. Case Presentation: Case 4, Figure 3 is not an ultrasound image. It is unclear what the blue arrows indicate. I do not understand why a pelvic image is necessary in this report.

Response: For case 4 (now case 3), Figure 3 has now been removed because it is irrelevant to the focus of this report.

  1. Case Presentation: Case 7, Figure 4 is not an ultrasound image. It is unclear why a pelvic image is required.

Response: For case 7 (now case 6), the pelvic ultrasound (Figure 6) has now been removed as well.

  1. Discussion and Conclusion: It is inappropriate to consider the mortality rate based on the course of only one case. In addition, I cannot agree with the claim in Case 5 that diagnostic endoscopy should be performed if the patient has diaphragm elevation in a patient whose general condition is stable. Diaphragm elevation is due to a mechanism different from diaphragm injury.

Response: Thank you for pointing this out. We acknowledge the mortality of one patient cannot be used as a generalized mortality. The mention of the case 1 mortality is an isolated instance and this has been addressed in the discussion session as well. For the second comment regarding case 5 (now case 4), the intervention was diagnostic laparoscopy (not endoscopy). Reference # 18 was used to back this up that diagnostic laparoscopy can be performed in stable patients.

Thanks again! We are willing to make any other necessary adjustments if needed. While an elevated diaphragm may not necessarily indicate a diaphragmatic injury, it can be a suggestive sign when considered in conjunction with other clinical signs and imaging findings.

Round 2

Reviewer 1 Report

Comments and Suggestions for Authors

The paper is suitable for publication in my opinion

Reviewer 2 Report

Comments and Suggestions for Authors

Thank you for correcting the points I pointed out. I understand the strength of this study compared to others is that it is a detailed case series of a single-center experience with blunt diaphragmatic injuries. However, this alone is not enough to make it superior to previous studies, and new findings derived from the results of this study need to be pursued to publish it as a paper. The following modifications could be attempted to make it more attractive.

Major Points

1.     As mentioned in the Conclusion, I think the main findings of this study are that “multiple injuries to the lower ribs of the 6th-12th ribs” or “high-energy injury mechanism” can be the keywords to suspect blunt diaphragmatic injury. I think these two points should be summarized and discussed primarily in the Discussion. The former is a particularly interesting finding. Other claims such as “cases with ISS of 24 or more have a poor prognosis” and “it is difficult to fully diagnose diaphragmatic injuries by Xp and CT alone, and diagnostic laparoscopy or surgery may be considered when necessary” are not actively supported by this study due to the small sample size. While these statements are certainly consistent with the results of previous studies, I think they are overestimated in this study.

2.     Case 4: As noted in the final paragraph of the Management and outcomes, diagnostic laparoscopy was performed in Case 4, but no diaphragmatic injury was identified. The presence of diaphragmatic elevation was used as the basis for the diagnosis of diaphragmatic injury, but it is possible that diaphragmatic elevation was present before the injury. I think it is inappropriate to include case 4 as a case for this study.

3.     Please describe the location of the rib fracture in Case 5-8.

4.     Management and outcomes: The contents of the first and second paragraphs overlap with the contents of the Introduction. I believe that these contents should be described in the Introduction.

5.     Discussions: The content of the last paragraph should be stated in the Introduction because it describes the problems of past studies and the points that this study intends to address.

6.     Conclusion: The recommendations for treatment strategies for blunt diaphragmatic injuries are overstated as a conclusion of this study. I think the author's assertion that there are limitations to this study and issues to be resolved in the future is valid.

7.     Table 1: The explanation should be included in the main text, and there is a lot of overlap with the main text, which seems redundant.

Minor Points

1.     Methods, last paragraph: The authors mention blunt traumatic dental injury (TDI), but I think “blunt traumatic diaphragm injury” is the correct term.

2.     Case 1, Figure 1: Please specify that image “A” was at the time of admission and image “B” was two days after admission.

Author Response

Dear reviewer,

Thank you so much for providing these comments. We have tried to respond to all of them. Please find point to point responses below: 

Major points

  1. As mentioned in the Conclusion, I think the main findings of this study are that “multiple injuries to the lower ribs of the 6th-12th ribs” or “high-energy injury mechanism” can be the keywords to suspect blunt diaphragmatic injury. I think these two points should be summarized and discussed primarily in the Discussion. The former is a particularly interesting finding. Other claims such as “cases with ISS of 24 or more have a poor prognosis” and “it is difficult to fully diagnose diaphragmatic injuries by Xp and CT alone, and diagnostic laparoscopy or surgery may be considered when necessary” are not actively supported by this study due to the small sample size. While these statements are certainly consistent with the results of previous studies, I think they are overestimated in this study.

Response: Thank you for highlighting this point. We have incorporated your suggestions and re-written the discussion to include:

One of the key findings of our study is the association between lower rib fractures (6th-12th ribs) and high-energy trauma as significant indicators of blunt TDI. In our cohort, 3 out of 7 patients had fractures to the lower ribs, and all but one sustained high-energy trauma (pedestrian strike), supporting the role of these factors in increasing the suspicion for DI.

However, while our findings suggest a potential association, the small sample size limits the ability to make definitive conclusions, and further research with larger cohorts is needed to confirm these trends. Additionally, although radiographic imaging was used in many of our cases, we found that CXR and CT were not always sufficient in diagnosing TDI, particularly in cases where no bowel contents were herniating into the thoracic cavity. This supports findings from previous studies indicating that alternative diagnostic methods, such as diagnostic laparoscopy or surgical exploration, may be necessary in patients with high clinical suspicion but negative initial imaging (11).

Regarding the ISS, while our case series suggests that patients with ISS greater than 24 tend to have poorer outcomes, it is important to acknowledge the limitations of our small sample size. A retrospective study by Cardoso et al. found similar findings, with mortality rates higher in patients with an ISS greater than 24 (19), but larger, multicenter studies are needed to validate the predictive value of ISS for TDI outcomes.

  1. Case 4: As noted in the final paragraph of the Management and outcomes, diagnostic laparoscopy was performed in Case 4, but no diaphragmatic injury was identified. The presence of diaphragmatic elevation was used as the basis for the diagnosis of diaphragmatic injury, but it is possible that diaphragmatic elevation was present before the injury. I think it is inappropriate to include case 4 as a case for this study.

Response: This case has now been entirely removed from our report (note: this has changed the numbering of our cases).

  1. Please describe the location of the rib fracture in Case 5-8.

Response: We have described the location of rib fractures.

Case 4: As per the autopsy report, the recorded location of rib fractures was right-sided ribs 3-7, right lateral ribs 3-5, right paravertebral ribs 10-12, and left anterior ribs 2-8.

Case 5: As per the autopsy report location of the rib fractures was noted as right-sided ribs 4, 6-8, and left lateral ribs 1-7

Case 6: We couldn’t find autopsy report for the previous patient so we replaced it with other case and add all relevant details based on autopsy report. This patient’s autopsy report clearly explains that there was blunt diaphragmatic injury along with multiple rib fractures but no location is described.

A 39-year-old Hispanic brown female with an unknown past medical history was brought by EMS as a pedestrian struck by a motor vehicle. It was RED trauma on prenotification with a reported GCS of 15. She showed loss of vital signs within 6 minutes of prehospital CPR. She was intubated in the trauma bay. ET tube was in place with condensation in the tube, and bilateral breath sounds were heard during bagging; however, no End-tidal carbon dioxide (EtCO2) was detected, presumably due to lack of circulation. There was a chest tube extending through the right sixth intercostal muscles and into the right pleural space. Associated with chest tube placement is an approximately 1-1/2″ minimally hemorrhagic incision through the right hemidiaphragm and into the right lobe of the liver. MTP was activated before the patient's arrival and resuscitative thoracotomy was performed. We requested patient data from the NTRACS Trauma Registry. At our center, the trauma registry records are updated based on Autopsy results. The diaphragm was significantly distended reporting intra-abdominal hemorrhage. Autopsy results confirmed that the patient had multiple rib fractures along with a blunt diaphragmatic injury. The reported ISS for this patient was 75.

Case 7: As per autopsy report displaced fracture of all anterior ribs bilaterally, lateral left ribs 2-12, posterior Left ribs 2-12, and posterior lateral Right ribs 2-12.

  1. Management and outcomes: The contents of the first and second paragraphs overlap with the contents of the Introduction. I believe that these contents should be described in the Introduction.

Response: Thank you for your valuable suggestions. We have incorporated the contents of the last two paragraphs from the management and outcome section into the introduction, as recommended. In doing so, we identified some redundancy and have streamlined the introduction to eliminated repetitive wordings. Similarly, we have removed the first two paragraphs from the management and outcome. The revised introduction now reads:

The diaphragm is a crucial skeletal muscle that separates the thoracic and abdominal cavities (1) and plays an essential role in activities like defecation and vomiting by regulating intra-thoracic and intra-abdominal pressure gradients (2). Embryologically, it develops from the septum transversum during week four of gestation and assumes a domelike appearance with a central tendinous part upon development (2). Anatomically, it attaches to the sternum, the sixth to twelfth ribs, the last thoracic vertebra, and the first three lumbar vertebrae (2). Despite its protection within the thoracic cage, the diaphragm can sustain traumatic injuries, either blunt or penetrating injury, through the thoracoabdominal region. Blunt injuries often result from motor vehicle accidents, falls from a height, or pedestrian strikes, whereas penetrating injuries include gunshot wounds, stab wounds, or sharp objects with the capacity to penetrate the thoracoabdominal partition. Injury to the diaphragm can also arise at birth, a term called congenital diaphragmatic hernia, which occurs due to a defect in the diaphragm during its development (3).

Diagnosing a blunt DI can be challenging because of its potential to remain undetected in a few cases, leading to delayed presentation with abdominal pain and respiratory symptoms several months after the acute event (9). Early diagnosis is crucial to prevent complications such as strangulation and perforation, which can arise if the injury is not promptly managed (12, 13).

Chest X-ray (CXR) and contrast-enhanced computed tomography (CT) are the primary imaging modalities for diagnosing TDI (8). While CXR is often the first imaging technique used in trauma patients, studies show that it is diagnostic in only about one-third of cases (10). On the other hand, CT is more sensitive and specific, detecting diaphragmatic injuries with a sensitivity of 71% and a specificity of 100%, particularly when abdominal contents are herniated into the thoracic cavity (10, 11). CT also provides detailed visualization of air or fluid in the thoracic cavity, helping to identify ruptures in the diaphragm (11). However, CT may miss injuries in cases where no bowel herniation is visible, making a high index of suspicion crucial (14).

Despite its higher sensitivity, CT is not infallible, and TDI may still be overlooked, especially when the injury is minute or when other associated injuries are more clinically apparent. A study found that in cases of left-sided blunt diaphragmatic ruptures, preoperative identification occurred in only 40-50% of cases, and a mere 0-10% for right-sided ruptures. Therefore, while CT is valuable, it may still require confirmation through surgical exploration, such as laparotomy or thoracotomy, in cases where clinical suspicion remains high.

The management of blunt TDI generally involves surgical intervention, with options including thoracotomy, laparotomy, or a combination of both, depending on the nature and severity of the injury. In stable patients with suspected TDI, diagnostic laparoscopy or thoracoscopy may be used to confirm the diagnosis (14, 15). For unstable patients or those with significant bowel herniation, immediate surgical exploration is warranted to prevent further complications.

The mortality rate related to diaphragmatic injuries can vary, ranging from 5% to 30% (4, 8-11). Right-sided TDI, while less common, carries an increased risk of mortality due to its association with other abdominal injuries (6). Studies have shown that early surgical intervention, particularly in younger patients, is associated with better survival rates (6, 14).

  1. Discussions: The content of the last paragraph should be stated in the Introduction because it describes the problems of past studies and the points that this study intends to address.

Response: Thank you for the suggestion. The content of the last paragraph in discussion has now been incorporated in the introduction.

“As most of the cases in our study died on arrival or arrived dead, we reviewed autopsy reports to confirm blunt TDI in such cases. We only included cases with more detailed information so that the severity of blunt TDI can be communicated to other researchers. This ensures the novelty of our study. Among the several published papers, very few groups have focused on writing a case series solely focusing on blunt TDI. Archives of published studies either include the majority of case reports with single cases or very few clinical trials. Not only did we refer to the patient’s medical chart, but we also referred to registry data to ensure exploration and detailed description of these cases wherever possible. At our center, out of 17 cases, except for a few, the rest were either dead on arrival or arrived dead due to injury severity or died within 2 days of admission. Our study is one of the novel studies proving the high mortality rate in patients with blunt TDI. Although the size of the studied patient population is small, it highlights the need to advance education programs to manage such life-changing incidents. The findings will impact trauma quality improvement practices regionally and nationally. Such crucial points impart novelty, educational value, and uniqueness to our study when compared to other reports.”

  1. Conclusion: The recommendations for treatment strategies for blunt diaphragmatic injuries are overstated as a conclusion of this study. I think the author's assertion that there are limitations to this study and issues to be resolved in the future is valid.

Response: Thank you for your thoughtful and thorough review of our case series. You raise a valid point, and we appreciate the time you have taken to provide feedback. We acknowledge that we did not clearly state earlier in the report that our recommendations were based on the findings from this case series and are observational in nature, rather than generalized treatment guidelines. We have now made this clarification in the report:

“These recommendations should be viewed as preliminary suggestions based on our case series and not definitive treatment strategies, given the small sample size and the observational nature of this study.

And we deleted:

“We call for the development of standardized trauma protocols that incorporate these recommendations, particularly in settings with limited resources. Protocols should be based on the mechanism of injury and clinical presentation, physical exam findings, and radiographic imaging ensuring that patients with potential TDI receive timely intervention.”

We believe it would be valuable to test these recommendations in larger, multicenter studies to determine their applicability and effectiveness. This would require a larger sample size and further research to validate these initial findings.

  1. Table 1: The explanation should be included in the main text, and there is a lot of overlap with the main text, which seems redundant.

Response: Thank you for pointing this out. We have now incorporated the explanation from table 1 into the main text. In doing so, we identified some redundancy in the main text and have removed repetitive details. For example, the following section was removed:

 “Case 4 had a CT scan without contrast showing right hemidiaphragm elevation without evidence of intraabdominal contents in the thorax. Although the patient was hemodynamically stable, an exploratory laparoscopy revealed no injury. This aligns with Al-Jehani et al., who mentioned a preference for the use of diagnostic laparoscopy in stable patients (18).

Although the patient in case 4 already lost pulse en route to the hospital, she underwent resuscitative thoracotomy in the trauma bay, which revealed an elevated diaphragm but did not survive her injury.

Overall, of our eight patients, three had CXR performed with one (33.3%) showing findings indicative of DI (case 3). Among the four patients who underwent CT imaging, DI was confirmed in two (50%) through the presence of herniated gastrointestinal (GI) content in the thorax (cases 1 and 3). However, CT failed to detect DI in the other two patients (cases 2 and 4), despite a high index of suspicion in case 4. This highlights that while CT is more sensitive and specific than chest radiographs, it may still miss injuries, especially when no visceral contents are present in the thoracic cavity, posing a challenge to early diagnosis.

In case 2, where CT scans failed to reveal diaphragmatic rupture, surgical exploration (VATS or thoracotomy) ultimately revealed the injuries. In case 4, where CT findings are inconclusive or absent, diagnostic laparoscopy and resuscitative thoracotomy helped confirm or rule out diaphragmatic tears.”

Minor points

  1. Methods, last paragraph: The authors mention blunt traumatic dental injury (TDI), but I think “blunt traumatic diaphragm injury” is the correct term.

  Response: Thank you for pointing out this error. We have corrected it, and we also noticed that 'blunt traumatic diaphragm injury' (abbreviated as 'blunt TDI' in our report) was mistakenly abbreviated as 'blunt traumatic dental injury' (TBI). To ensure consistency, we have made the necessary corrections and standardized the abbreviation to 'TDI' for traumatic diaphragm injury throughout the document."

  1. Case 1, Figure 1: Please specify that image “A” was at the time of admission and image “B” was two days after admission.

Response: This has now been addressed. In Figure 1, Image A is now labeled 'at the time of admission' in parenthesis following the CT scan, and Image B is now labeled 'two days after admission' in parenthesis as well following the CTA."

Thank you for the time taken to provide a wonderful comments and feedback on this report. We hope our responses addresses your questions. We are willingly to make any other necessary adjustment if needed.

Sincerely,

Bharti Sharma (Corresponding author)

Round 3

Reviewer 2 Report

Comments and Suggestions for Authors

Thank you again for correcting the points I pointed out. The arguments have become clearer and the author's point has become easier to understand. To further improve the article, I would like you to correct the following points.

1.     Abstract: There is no mention of rib fractures, one of the main points of discussion, and this is dissociated from the content of the revised paper. The claim that diaphragmatic injuries can also occur in low-energy incidents is not mentioned in the main text, so it is inappropriate to mention it here. In fact, in the seven cases presented, all the injuries were sustained with a high-energy injury mechanism. Please revise the Abstract appropriately based on the Conclusion of the revised paper.

2.     Introduction: The final paragraph is a direct copy and pasted version of the previous revised paper, but even if you write this before the case presentation, it is difficult to understand because there is no context. Please write an appropriate Introduction that is consistent with the surrounding sentences.

3.     Please describe the location of the rib fracture in Case 3.

4.     Please note that in Case 6, there were multiple rib fractures, but the location of the rib fractures was unknown. Please note that this was a fatal case. Since this is not an M&M conference, I don't think it is necessary to mention endotracheal intubation.

5.     The content of “Management and Outcomes” is a summary of the results of this study, so a comparison with previous literature should not be made here, but in the Discussion. It will be easier to read if you include the contents of “Management and Outcomes” and the “first paragraph of the Discussion” at the end of the Case Presentation. It is also better to include Table 1 at the end of the Case Presentation.

6.     Discussion: The second paragraph states, “One of the key findings of our study is the association between lower rib fractures (1st-12th ribs) …” but I think “lower rib fractures (6th-12th ribs)” is correct.

7.     Table 1 is a summary of the results, so it would be better to include it at the end of the "Case Presentation". Also, please add information about the location of the rib fracture to Table 1. "Comments" are matters to be discussed in the Discussion, so there is no need to include them in this table. Please update Case 6 with new information.

Author Response

Dear reviewer,

Thank you so much for providing these comments. We have tried to respond to all of them. Please find point to point responses below

Comments:

  1. Abstract: There is no mention of rib fractures, one of the main points of discussion, and this is dissociated from the content of the revised paper. The claim that diaphragmatic injuries can also occur in low-energy incidents is not mentioned in the main text, so it is inappropriate to mention it here. In fact, in the seven cases presented, all the injuries were sustained with a high-energy injury mechanism. Please revise the Abstract appropriately based on the Conclusion of the revised paper.

Response: Thanks for pointing this out. We have removed the statement claiming diaphragmatic injuries can also occur in low-energy incidents. We have included statement about the seven cases presented that all the injuries were sustained with a high-energy injury mechanism. We have revised the entire abstract and maintained the word limit. Please see the following:

Introduction: Detection of blunt traumatic diaphragm injury (TDI) can be challenging in the absence of surgical exploration. Our objective is to study the mechanisms of injury and detection modes for patients with blunt TDI. Methods: This is a single-center, retrospective review conducted in a level 1 trauma center, 2016 to 2023, inclusive. We identified seven patients with blunt TDI using the primary mechanisms and trauma type. Results: Out of 7 patients, 2 were associated with motor vehicle collisions, 4 were pedestrians struck, and 1 fell from the stairs. The mean ISS was 48.4 (29-75). Of the seven patients with blunt TDI, four died in the trauma bay – two from traumatic arrest, and two died spontaneously. Multiple rib fractures were one of the common injury patterns in 6 cases whereas in remaining 1 case blunt TDI was confirmed was confirmed at laparotomy and repaired. One patient died two days after admission. Of the two patients who survived, one had a TDI identified during video-assisted thoracic surgery (VATS) for retained hemothorax, and one patient had a TDI repaired during emergent exploratory laparotomy for other injuries. In the remaining 4 patients blunt TDI was confirmed based on their autopsy reports. Conclusions: Injuries in all 7 cases were sustained with a high-energy injury mechanism. Multiple rib fractures were reported in 6 cases. Imaging can be useful in detecting such injuries but is not 100% sensitive. A high index of suspicion should be maintained for DI in high-energy blunt trauma.

  1. Introduction: The final paragraph is a direct copy and pasted version of the previous revised paper, but even if you write this before the case presentation, it is difficult to understand because there is no context. Please write an appropriate Introduction that is consistent with the surrounding sentences.

Response: Thanks, and yes, we have revised the introduction section to maintain consistency

The diaphragm is a crucial skeletal muscle that separates the thoracic and abdominal cavities (1) and plays an essential role in activities like defecation and vomiting by regulating intra-thoracic and intra-abdominal pressure gradients (2). Embryologically, it develops from the septum transversum during week four of gestation and assumes a domelike appearance with a central tendinous part upon development (2). Anatomically, it attaches to the sternum, the sixth to twelfth ribs, the last thoracic vertebra, and the first three lumbar vertebrae (2). Despite its protection within the thoracic cage, the diaphragm can sustain traumatic injuries, either blunt or penetrating injury, through the thoracoabdominal region. Blunt injuries often result from motor vehicle accidents, falls from a height, or pedestrian strikes, whereas penetrating injuries include gunshot wounds, stab wounds, or sharp objects with the capacity to penetrate the thoracoabdominal partition. Injury to the diaphragm can also arise at birth, a term called congenital diaphragmatic hernia, which occurs due to a defect in the diaphragm during its development (3).

Diagnosing a blunt TDI can be challenging because of its potential to remain undetected in a few cases, leading to delayed presentation with abdominal pain and respiratory symptoms several months after the acute event (4). Early diagnosis is crucial to prevent complications such as strangulation and perforation, which can arise if the injury is not promptly managed (5-6). Chest X-ray (CXR) and contrast-enhanced computed tomography (CT) are the primary imaging modalities for diagnosing TDI (7). While CXR is often the first imaging technique used in trauma patients, studies show that it is diagnostic in only about one-third of cases (8).

On the other hand, CT is more sensitive and specific, detecting DI with a sensitivity of 71% and a specificity of 100%, particularly when abdominal contents are herniated into the thoracic cavity (8-9). CT also provides detailed visualization of air or fluid in the thoracic cavity, helping to identify ruptures in the diaphragm (9). Despite its higher sensitivity, CT is not infallible, and TDI may still be overlooked, especially when the injury is minute or when other associated injuries are more clinically apparent (10). In cases focusing on left-sided blunt diaphragmatic ruptures, preoperative identification has been reported to occur in only 40-50% of cases, and a mere 0-10% for right-sided ruptures. Therefore, while CT is valuable, it may still require confirmation through surgical exploration, such as laparotomy or thoracotomy, in cases where clinical suspicion remains high.

Blunt TDI is uncommon and there are not many reported studies focusing on blunt TDI. Hence, with an objective to analyze various methods of detecting blunt TDI, the mechanism of injury, and the role of the Injury Severity Score (ISS) in patient outcomes, we bring forth this case series. We also reviewed patient demographics. This case series included 7 patients with blunt TDI from 2016 to 2023, inclusive. The demographic breakdown included 1 White American, 1 Black American, 2 Asians, and 3 from other racial backgrounds, with a mean age of 47.1 years (range 28-88 years). Further details on all included cases is elaborated in the following sections.

  1. Please describe the location of the rib fracture in Case 3.

Response: For this case, we didn’t find any reported rib fracture in patient’s chart. CXR and CT-A/P were consistent with left TDI.  Blunt traumatic diaphragmatic injury was confirmed at laparotomy and repair

  1. Please note that in Case 6, there were multiple rib fractures, but the location of the rib fractures was unknown. Please note that this was a fatal case. Since this is not an M&M conference, I don't think it is necessary to mention endotracheal intubation.

Response: Information on endotracheal intubation has been removed. We have added the location of the rib fractures. Autopsy results confirmed that the patient had a blunt diaphragmatic injury along with multiple hemorrhagic fractures of the left lateral ribs 1-7, anterior right rib 4 and 6-8.

  1. The content of “Management and Outcomes” is a summary of the results of this study, so a comparison with previous literature should not be made here, but in the Discussion. It will be easier to read if you include the contents of “Management and Outcomes” and the “first paragraph of the Discussion” at the end of the Case Presentation. It is also better to include Table 1 at the end of the Case Presentation.

Response: We agree. We have rearranged the information as per your guidance. We have included first paragraph of the Discussion into Management and Outcomes after case presentation.

  1. Management and Outcome

Diagnosing DI after a blunt trauma can be challenging due to its potential to remain undetected leading to delayed presentation years after the initial injury (16, 17). The management of blunt TDI generally involves surgical intervention, with options including thoracotomy, laparotomy, or a combination of both, depending on the nature and severity of the injury. In stable patients with suspected TDI, diagnostic laparoscopy or thoracoscopy may be used to confirm the diagnosis (11-12). For unstable patients or those with significant bowel herniation, immediate surgical exploration is warranted to prevent further complications (4). The mortality rate related to DI can vary, ranging from 5% to 30% (13, 14-17). Right-sided TDI, while less common, carries an increased risk of mortality due to its association with other abdominal injuries (18). Studies have shown that early surgical intervention, particularly in younger patients, is associated with better survival rates (12, 18).

Our case series aimed to analyze the methods of detecting blunt TDI, the mechanism of injury, and the role of the Injury Severity Score (ISS) in patient outcomes. We also reviewed patient demographics. This case series included 7 patients from 2016 to 2023 with blunt TDI. The demographic breakdown included 1 White American, 1 Black American, 2 Asians, and 3 from other racial backgrounds, with a mean age of 47.1 years (range 28-88 years).

In our study, case 1 had DI detected on a chest CT scan, but the patient declined surgical intervention opting for comfort care and passing away the following day. This finding is consistent with literature indicating that right hemidiaphragm injury has an increased risk of mortality (18). However, it is important to note that this case alone should not be used to generalize mortality rates for all patients with blunt TDI. The mortality observed in case 1 is an isolated instance and cannot be considered representative of the broader outcomes in the series.

In case 2, the patient underwent video-assisted thoracotomy surgery (VATS) for rib repair, and a DI was noted on the left, which was repaired with a 0-ethibond stitch, leading to a favorable outcome. In case 3, blunt TDI was detected on exploratory laparotomy with repair using a 2-0 ethibond suture. The patient survived. In case 4, resuscitative thoracotomy in the trauma bay revealed a distended diaphragm, but the patient succumbed to his injuries. Cases 5, 6, and 7 died from traumatic arrest either before arriving at the ED or were pronounced dead in the trauma bay without any CT imaging performed to indicate DI before their death.

  1. Discussion: The second paragraph states, “One of the key findings of our study is the association between lower rib fractures (1st-12th ribs) …” but I think “lower rib fractures (6th-12th ribs)” is correct.

Response: Yes, we agree. We have incorporated this change.

  1. Table 1 is a summary of the results, so it would be better to include it at the end of the "Case Presentation". Also, please add information about the location of the rib fracture to Table 1. "Comments" are matters to be discussed in the Discussion, so there is no need to include them in this table. Please update Case 6 with new information.

Response: We have included table 1 at the end of case presentation. Also, added location of rib fracture for all cases (wherever applicable). Updated information for case 6.

Case No.

Mechanism of injury

Patient

demographics

Age

Gender

CT scan findings

Surgical intervention

Outcome

ISS

Location of rib fracture

1

Mechanical fall at home

White American

88

Female

Abdominal contents in right hemidiaphragm

Comfort care (no surgery)

Died next day

29

anterior right 3-5 ribs, right 8-10 ribs, and right 6-7 ribs

2

Motor vehicle accident

non-Hispanic

30

Male

Mildly displaced fracture of left ribs 6th – 12th

VATS and thoracotomy for rib repair, diaphragm repair

survived

22

left 6 -12 ribs

3

Pedestrian strike

Asian

28

Female

Abdominal content in left hemithorax

Exploratory laparotomy, diaphragm repair

survived

41

No rib fracture reported, blunt TDI was confirmed at laparotomy and repaired

4

Pedestrian strike

Hispanic

36

Female

Elevated diaphragm (no CT scan)

Resuscitative thoracotomy

died

43

right-sided ribs 3-7, right lateral ribs 3-5, right paravertebral ribs 10-12, and left anterior ribs 2-8

5

Pedestrian strike

Asian

68

Female

No CT scan performed

No surgical intervention

died

54

right-sided ribs 4, 6-8, and left lateral ribs 1-7

6

Motor vehicle accident

Hispanic

48

Male

No CT scan performed

No surgical intervention

Died

75

multiple hemorrhagic fractures of the left lateral ribs 1-7, anterior right rib 4 and 6-8.

7

Pedestrian strike

Black American

32

Male

No CT scan performed

Died without surgical intervention

died

75

all anterior ribs bilaterally, lateral left ribs 2-12, posterior Left ribs 2-12, and posterior lateral Right ribs 2-12

Thank you for providing such helpful comments and feedback on this paper. We hope our responses addresses your questions. We are willing to make any other necessary adjustment if needed.

Sincerely,

Bharti Sharma (Corresponding author)

Round 4

Reviewer 2 Report

Comments and Suggestions for Authors

Some parts may seem redundant, I think that the argument of the paper has become clearer and easier for readers to understand.  Good job.